# Baseline Circulating Blood Cell Counts and Ratios and Changes Therein for Predicting Immune-Related Adverse Events during Immune Checkpoint Inhibitor Therapy: A Multicenter, Prospective, Observational, Pan-Cancer Cohort Study with a Gender Perspective

**DOI:** 10.3390/cancers16010151

**Published:** 2023-12-28

**Authors:** Lucía Teijeira, Mireia Martínez, Amaia Moreno, Ibone de Elejoste, Berta Ibáñez-Beroiz, Virginia Arrazubi, Isabela Díaz de Corcuera, Iñaki Elejalde, Ana Campillo-Calatayud, Iñigo Les

**Affiliations:** 1Servicio de Oncología Médica, Hospital Universitario de Navarra, 31008 Pamplona, Spain; lucia.teijeira.sanchez@navarra.es (L.T.); virginia.arrazubi.arrula@navarra.es (V.A.); 2Servicio de Oncología Médica, Hospital Universitario Araba, Servicio Vasco de Salud Osakidetza, 01009 Vitoria-Gasteiz, Spain; mireia.martinezkareaga@osakidetza.eus; 3Grupo de Investigación en Cáncer de Pulmón, Instituto de Investigación Sanitaria Bioaraba, 01009 Vitoria-Gasteiz, Spain; 4Servicio de Oncología Médica, Hospital Universitario Galdakao, 48960 Galdácano, Spain; amaia.morenopaul@osakidetza.eus (A.M.); isabela.diazdecorcuerafrutos@osakidetza.eus (I.D.d.C.); 5Servicio de Oncología Médica, Hospital Universitario Donostia, 20014 San Sebastián, Spain; ibone.deelejosteechebarria@osakidetza.eus; 6Servicio de Metodología, IdISNA, Navarrabiomed-Universidad Pública de Navarra, 31008 Pamplona, Spain; berta.ibanez.beroiz@navarra.es; 7Servicio de Medicina Interna, Hospital Universitario de Navarra, 31008 Pamplona, Spain; ji.elejalde.guerra@navarra.es; 8Unidad de Enfermedades Autoinmunes Sistémicas, Servicio de Medicina Interna, Hospital Universitario de Navarra, 31008 Pamplona, Spain; 9Grupo de Enfermedades Inflamatorias e Inmunomediadas, IdISNA, Navarrabiomed-Universidad Pública de Navarra, 31008 Pamplona, Spain; ana.campillo.calatayud@navarra.es; 10Departamento de Ciencias de la Salud, Universidad Pública de Navarra, 31008 Pamplona, Spain

**Keywords:** immune checkpoint inhibitors, immune-related adverse events, gender perspective, cancer, prediction, blood cell counts, blood cell ratios

## Abstract

**Simple Summary:**

The growing use of immune checkpoint inhibitors (ICIs) for the treatment of patients with solid tumors has led to a proportional increase in the incidence of toxic effects in the form of immune-related adverse events (irAEs). In this study, we show that two readily available blood cell parameters, namely, high absolute lymphocyte count before ICI initiation and early decline in absolute neutrophil count after ICI initiation, can predict irAEs during follow-up. Interestingly, however, the predictive ability of both pre-ICI absolute lymphocyte count and post-ICI absolute neutrophil count differ significantly between men and women. In the prediction of irAEs, we also describe an interaction between female gender and a decrease in absolute neutrophil count after the first cycle of ICI therapy. These findings should lead to the development of new predictive models for irAEs that are able to capture sex-related differences in ICI-induced toxicity.

**Abstract:**

Several factors have been associated with the occurrence of immune-related adverse events (irAEs) induced by immune checkpoint inhibitor (ICI) therapy. Despite their availability, the predictive value of circulating blood cell parameters remains underexplored. Our aim was to investigate whether baseline values of and early changes in absolute neutrophil count (ANC), absolute lymphocyte count (ALC), other blood cell counts, and lymphocyte-related ratios can predict irAEs and whether sex may differentially influence this potential predictive ability. Of the 145 patients included, 52 patients (35.8%) experienced at least one irAE, with a 1-year cumulative incidence of 41.6%. Using Fine and Gray competing risk models, we identified female sex (hazard ratio (HR) = 2.17, 95% confidence interval (CI) = 1.20–3.85), high ALC before ICI initiation (HR = 1.63, 95% CI = 1.09–2.45), and low ANC after ICI initiation (HR = 0.81, 95% CI = 0.69–0.96) as predictors of irAEs. However, ALC and ANC may only have an impact on the risk of irAEs in women (stratified for female sex, ALC-related HR = 2.61, 95% CI = 1.40–4.86 and ANC-related HR = 0.57, 95% CI = 0.41–0.81). Priority should be given to developing models to predict ICI-related toxicity and their validation in various settings, and such models should assess the impact of patient sex on the risk of toxicity.

## 1. Introduction

Immunotherapy is considered a breakthrough in cancer treatment. The most widely used type of immunotherapy involves immune checkpoint inhibitors (ICIs), which block inhibitory receptors in the immune system, such as the cytotoxic T-lymphocyte-associated antigen-4 (CTLA-4), programmed cell death protein-1 (PD-1), and PD-1 ligand 1. Nowadays, ICIs are increasingly used as frontline therapy in patients with solid organ cancer. By blocking certain suppressive pathways, ICIs promote T-cell activation, which induces tumor cell death but can also trigger so-called immune-related adverse events (irAEs) [1] that may mimic or exacerbate autoimmune diseases. Despite pathogenic and clinical differences, all ICIs can cause this well-described unique toxicity profile. Given the high incidence (up to 50% in patients treated with ICIs), the potential risk of toxicity-related damage, and correlation with ICI response rate, irAEs impact the quality of life and prognosis of a substantial proportion of patients with cancer [2]. Therefore, pragmatic and validated predictors of irAEs are urgently needed in clinical practice.

Though not yet validated, a wide variety of biomarkers have been proposed to predict irAEs [3], including circulating blood cell counts obtained from routine blood tests. For clinicians, the ready availability, low cost, and ease of interpretation of various blood cell counts for the prediction or early detection of irAEs may be of great interest. There is already evidence suggesting that certain blood cell parameters, such as baseline absolute neutrophil, lymphocyte, monocyte, and eosinophil counts; baseline platelet count; and increases in leukocyte, lymphocyte, and eosinophil counts during follow-up, are associated with a higher risk of irAEs [4,5,6]. There are also several blood cell ratios, the most common being the neutrophil-to-lymphocyte ratio (NLR), which may help predict irAEs before and after ICI initiation, although previous studies have yielded inconsistent results [7,8].

Further, to date, most studies on circulating blood cell counts and ratios have suffered from the same limitations as studies on other biomarkers of immune-related toxicity, namely, a retrospective design; limited time frame (usually considering only baseline data or short follow-up periods); and restriction to a specific type of cancer, irAE, or ICI agent. In particular, few prospective studies have evaluated the clinical value of fluctuations in blood cell counts and ratios for predicting irAEs in pan-cancer cohorts in the long term.

On the other hand, the potential influence of patient sex on the risk of developing irAEs remains a controversial issue in the literature. It is generally accepted that women have a more effective innate and adaptive immune response than men [9]. This apparently beneficial phenomenon results in an increased susceptibility to autoimmune and inflammatory diseases [10]. Based on our previous experience [11], women may be more predisposed to experiencing irAEs, but recent studies have provided conflicting data in this regard [12,13]. We believe that patient sex is a differentiating factor that has traditionally been overlooked in biomedical research. These contradictions and knowledge gaps warrant the adoption of a gendered perspective in research in this field.

In this context, we hypothesized that baseline blood counts and ratios and changes in these values early in the course of ICI therapy may predict the occurrence of irAEs in patients who are given this type of therapy and that patient sex may interact with these potentially predictive blood parameters. Therefore, the aim of this study was to prospectively investigate the value of peripheral blood cell counts and ratios for predicting irAEs in a pan-cancer cohort of patients treated with ICIs with a gender perspective.

## 2. Material and Methods

### 2.1. Study Design and Participants

This study consisted of a preliminary analysis of data from the AUTENTIC project, a multicenter observational prospective cohort study collecting samples from patients receiving ICIs for cancer treatment. The AUTENTIC study focuses on autoantibodies presumed to be predictive of irAEs [14], and it was designed by taking into account the REporting recommendations for tumor MARKer prognostic studies (REMARK).

All patients diagnosed with solid organ cancer amenable to ICI therapy according to current guidelines were considered eligible for this study, and the sample was obtained by recruiting consecutive cases. Patients were enrolled by medical oncologists in the outpatient clinics of four tertiary hospitals: Hospital Universitario de Navarra (Pamplona), Hospital Universitario Araba (Vitoria), Hospital Universitario Galdakao (Galdácano), and Hospital Universitario Donostia (San Sebastián).

Participants were required to meet the following inclusion criteria: (1) initiation of treatment with one ICI or a combination of ICIs and (2) ICI-naïve status (though patients were allowed to have received other systemic therapies for cancer, such as chemotherapy or tyrosine-kinase inhibitors). Patients were excluded if they met any of the following exclusion criteria: (1) life expectancy of less than 3 months from the initiation of ICI therapy; (2) any contraindication to ICI therapy, such as active severe autoimmune disease or poor performance status as assessed by an Eastern Cooperative Oncology Group (ECOG) score ≥3; (3) concurrent combination treatment with chemotherapy, tyrosine kinase inhibitors, or other specific cancer therapy; or (4) active immunosuppressive treatment, including prednisone at doses >10 mg/day or equivalent.

The ICIs used, namely, anti-PD-1 antibodies (nivolumab, pembrolizumab, and cemiplimab), anti-PD-L1 antibodies (atezolizumab, durvalumab, and avelumab), and anti-CTLA-4 antibodies (ipilimumab), and their tumor indications at the time of the design of this study are detailed in the Appendix A.

### 2.2. Procedures and Variables

The full AUTENTIC protocol has been published previously [14]. In brief, once enrolled in the study, patients were monitored in accordance with current clinical practice guidelines [15,16]. The follow-up intervals were dependent on the treatment schedule of each specific ICI and the occurrence of complications as well as the discretion of the attending physician. For the purposes of the current study, two blood samples were obtained: a baseline sample taken before ICI initiation and a follow-up sample taken just before the second ICI cycle, hereafter referred to as “pre-first ICI cycle” and “post-first ICI cycle” samples, respectively. The post-first ICI cycle sample was obtained 2 or 3 weeks after the first ICI cycle in line with the timing of the administration of the second ICI dose in accordance with the summaries of product characteristics used in the cohort. Patients were censored if they had an irAE, experienced cancer progression, or died.

The dependent variable was the occurrence of an irAE, defined as any sign, symptom, syndrome, or disease resulting from an immune-mediated mechanism during the treatment period with an ICI once other causes, including tumor progression, had been ruled out. Other toxicity-related variables prospectively recorded were the irAE type and grade according to the Common Terminology Criteria for Adverse Events v. 5.0 [17]. As exposure variables, we included the following blood cell counts: white blood cell count (WBC), absolute neutrophil count (ANC), absolute lymphocyte count (ALC), absolute monocyte count (AMC), absolute eosinophil count (AEC), and platelet count (PC). Using these cell counts, we estimated the following blood cell ratios: NLR (calculated as ANC/ALC), derived NLR (calculated as ANC/(WBC − ALC)), monocyte-to-lymphocyte ratio (MLR, calculated as AMC/ALC), eosinophil-to-lymphocyte ratio (ELR, calculated as AEC/ALC), and platelet-to-lymphocyte ratio (PLR, calculated as PC/ALC). The cell counts were obtained by automated methods in three of the four participating hospitals using an Abbott^®^ Alinity hq hematology analyzer based on advanced multiangle polarized scatter separation technology and in the fourth hospital (Hospital Universitario de Navarra) using a Beckman Coulter^®^ DxH 900 hematology analyzer based on volume, conductivity, and scatter technology.

Data were also gathered on other potential exposure variables, collectively referred to as patient-related characteristics, for inclusion in the predictive model: demographic features (age and sex), clinical data (body mass index, site of the primary tumor, and pre-existing autoimmune diseases), laboratory parameters other than blood cell counts and ratios (baseline glomerular filtration rate), and therapeutic modality (mono or dual (i.e., ipilimumab plus nivolumab) therapy with ICIs).

### 2.3. Objectives

The main objective of the study was to assess whether values of blood cell counts and ratios at baseline (i.e., pre-first ICI cycle) and early dynamic changes in these values (i.e., post-first ICI cycle) are predictive of the risk of developing irAEs in cancer patients treated with ICIs. The primary endpoint was the cumulative incidence of a first irAE during the follow-up. Given the design of the study, which focused on identifying biomarkers with predictive value prior to the development of irAEs, in patients who experienced more than one irAE, only the first irAE was included in the analysis.

The specific objectives were: (1) to describe patient-related characteristics, irAE type, and cumulative incidence of irAEs to (2) analyze the magnitude and significance of changes in blood cell parameters and the relationship between these changes and other patient-related characteristics; (3) estimate the association of patient-related characteristics and blood cell parameters with the risk of developing irAEs; and (4) assess the influence of patient sex on the risk of developing irAEs, allowing possible interactions between factors.

### 2.4. Statistical Analysis

Quantitative variables were expressed using means with standard deviations or medians with ranges, as appropriate, and presented with error bar plots, whereas categorical variables were expressed as frequencies with percentages. Cumulative incidence over time was assessed using a Fine and Gray competing risk method [18].

To analyze the magnitude and significance of changes in blood cell counts and ratios, intraindividual comparisons of these parameters before and after the first ICI cycle (i.e., pre- and post-first ICI cycle values, respectively) were performed using the Wilcoxon signed-rank test for paired data.

To explore the association of candidate predictive variables with the risk of experiencing an irAE, univariate Fine and Gray competing risk models were fitted, with an irAE as the main event, death as the competing event, and each patient-related characteristic as an exposure variable. The exposure variables considered were sex, pre-existing autoimmune disease, mono or dual therapy with ICIs, and each blood cell count and ratio. A multivariate Fine and Gray competing risk model was finally fitted including all significant variables. In addition, an interaction term was included to assess the interaction between sex and these variables and whether the effect of blood cell counts and ratios on irAE occurrence differed by sex; when significant, sex-stratified analysis was performed.

The R software (version 4.3.1, R Foundation, Vienna, Austria) and IBM SPSS Statistics for Windows (version 28.0.1.1, Armonk, NY, USA: IBM Corp.) were used for this analysis.

### 2.5. Ethical Considerations

The study was conducted in accordance with the International Council for Harmonization Guidelines for Good Clinical Practice E6 (R2), the principles of the Declaration of Helsinki, and local regulations. Informed consent was obtained from all participants before their enrollment. The study protocol was reviewed and approved by the Spanish Agency of Medicines and Medical Devices (code: ILB-NIV-2018-01), the Research Ethics Committee of the Basque Country (code: PI2018106 (EPA-SP)), and local ethics committees of each of the participating hospitals. The study protocol has been registered and is available on ClinicalTrials.gov (NCT03868046) [14].

## 3. Results

### 3.1. Description of the Cohort and Immune-Related Adverse Events

Table 1 summarizes the baseline patient characteristics and treatment modalities used in the cohort. Briefly, 145 patients (overall mean age of 65.8 ± 9.7 years; 112 men (77.2%)) were enrolled in the study and followed up for a median of 82 (range 6–855) days from the time of ICI initiation. A total of 9 patients (6.2%) had a previous diagnosis of autoimmune disease, the most common being psoriasis (4 cases). The primary cancer was non-small-cell lung cancer in 50 (34.5%) cases, renal cell carcinoma in 28 (19.3%), head and neck squamous cell carcinoma in 25 (17.2%), urothelial carcinoma in 18 (12.4%), and melanoma in 14 (9.7%). Other types of cancer were less frequent in our series, with just four cases of gastric adenocarcinoma, two of colorectal adenocarcinoma, two of malignant pleural mesothelioma, and one each of pancreatic adenocarcinoma and Merkel cell carcinoma.

During follow-up, 52 patients (35.8%) developed at least one irAE (Table 2). Notably, 49 of these 52 irAEs (94.5%) were grade 1 or 2, and the other 3 (5.5%) were grade 3. The cumulative irAE incidence at 1 year in the presence of death as a competing risk was 41.6% (95% confidence interval (CI), 32.6–50.4%), with a median time to first irAE of 47.5 (range 10–212) days after ICI initiation (Figure 1). The differences in the cumulative irAE incidence over time using a Kaplan–Meier method and a Fine and Gray competing risk method are shown in Appendix A. The most common types of first irAEs were cutaneous and thyroid disorders (Table 2), accounting for 57.7% of all irAEs in our cohort. Appendix A summarizes the subsequent irAEs documented in the cohort, that is, after the first irAE, which was considered a censoring event for this study.

### 3.2. Baseline and Follow-Up Blood Cell Counts and Ratios

For the 145 patients included in the study, pre- and post-first ICI cycle samples were obtained in 145 (100%) and 134 (92.4%) cases, respectively. In all 11 cases in which post-first ICI cycle samples were not included in the analysis, the reason was patient censoring before reaching the second ICI cycle due to death in 7 cases and cancer progression or development of an irAE in 2 cases each.

Appendix A lists all the blood cell parameters under study at baseline (pre-first ICI cycle) and after the first ICI cycle (post-first ICI cycle) and their relative increase between pre- and post-first ICI cycle sampling in the 134 patients who reached the second ICI cycle without being censored. For these 134 patients, Figure 2 plots the pre- and post-first ICI cycle values of WBC, ANC, and ALC, which were the parameters found to be relevant for predicting adverse events in subsequent analysis (see below), while pre-and post-therapy values of the other blood cell parameters (AMC, AEC, PC, NLR, dNLR, MLR, ELR, and PLR) are presented in Appendix A.

### 3.3. Relationship between Blood Cell Parameters and Other Patient-Related Variables

Appendix A summarizes the values of all the blood cell parameters under study (WBC, ANC, ALC, AMC, AEC, PC, NLR, dNLR, MLR, ELR, and PLR) at baseline (pre-first ICI cycle) and after the first ICI cycle (post-first ICI cycle) by patient characteristics (age, sex, pre-existing autoimmune disease, and mono or dual therapy with ICIs) in the 134 patients not censored between the first and second ICI cycle. For these 134 patients, the pre- and post-first ICI cycle WBC, ANC, and ALC are plotted in Appendix A by patient characteristics (age, sex, pre-existing autoimmune disease, ICI monotherapy or dual therapy).

### 3.4. Factors Associated with Immune-Related Adverse Events

Table 3 summarizes the factors associated with the occurrence of irAEs in our cohort. The univariate Fine and Gray competing risk models showed that female sex and dual therapy with ipilimumab plus nivolumab were associated with a higher risk of developing an irAE (hazard ratio (HR) = 2.04, 95% CI = 1.10–3.70, *p* = 0.025 for women; HR = 1.87, 95% CI 1.02–3.43, *p* = 0.043 for patients receiving dual ICI therapy). Regarding blood cell parameters, a high ALC before the first ICI cycle (HR = 1.60, 95% CI = 1.11–2.31, *p* = 0.011) and a low ANC after the first ICI cycle (HR = 0.81, 95% CI = 0.68–0.95, *p* = 0.012) predicted irAEs during follow-up. The effect of other potentially explanatory variables is detailed in Appendix A.

When these four factors (sex, dual ICI therapy, pre-first cycle ALC, and post-first cycle ANC) were included in a multivariate Fine and Gray competing risk model, only female sex (HR = 2.17, 95% CI = 1.20–3.85, *p* = 0.010), high pre-first ICI cycle ALC (HR = 1.63, 95% CI = 1.09–2.45, *p* = 0.018), and low post-first ICI cycle ANC (HR = 0.81, 95% CI = 0.69–0.96, *p* = 0.015) remained significant risk factors for irAEs (Table 3). The results of the multivariate Fine and Gray competing risk model did not change when the analysis was adjusted for dual ICI therapy.

### 3.5. Interaction between Patient Sex and Blood Cell Parameters

We found a significant interaction between patient sex and post-first ICI cycle ANC (HR = 1.56, 95% CI = 1.04–2.33, *p* = 0.031) but not between patient sex and pre-first ICI cycle ALC (HR = 0.54, 95% CI = 0.23–1.25, *p* = 0.150). Given these findings, we ran a multivariate Fine and Gray competing risk model stratified by sex, and interestingly, pre-first ICI cycle ALC and post-first ICI cycle ANC were both found to have predictive value in women but not in men treated with ICIs (Table 4). Figure 3 shows the cumulative incidence of irAEs as a function of patient sex and predefined cut-offs based on previously published data on pre-first ICI cycle ALC (cut-off: 2.0 thousand cells per microliter (K/µL)) [4] and post-first ICI cycle ANC (cut-off: 4.0 K/µL) [5].

## 4. Discussion

In this prospective study, we found that female sex, high pre-first ICI cycle ALC, and low post-first ICI cycle ANC could predict the occurrence of irAEs in cancer patients treated with ICIs. Interestingly, pre-first ICI cycle ALC and post-first ICI cycle ANC were both found to have predictive value in women but not in men. In addition, we found a significant interaction between female sex and post-first ICI cycle ANC in that the risk of an irAE increased further when female sex was combined with a low post-first ICI cycle ANC.

Although irAEs in our series were less severe, the cumulative irAE incidence was 41.6%, similar to rates previously reported [19,20]. As expected, immune-mediated toxicity was more common and more severe and potentially serious in patients treated with CTLA-4/PD-1 inhibitor combination therapy than in patients treated with an ICI monotherapy [21,22]. Nonetheless, very severe irAEs can also occur with monotherapy [7]. A rapid onset of toxic manifestations from the start of ICI therapy and systemic and/or myocardial involvement may indicate the development of high-grade irAEs [23,24]. In our study, dual ICI therapy was associated with an almost twofold risk of irAEs compared to monotherapy in the univariate analysis but not in the multivariate model (Table 3), probably because of the small number of patients treated with dual ICI therapy. Nevertheless, given the clinical relevance of high-grade toxicity, ICI combination therapy should be used as a potential predictor of irAEs.

Regarding blood cell parameters, patients with high baseline ALC had an increased risk of developing an irAE compared to patients with low baseline ALC. An association between high baseline values of ALC and ICI-related toxicity has been described previously by several authors, including Diehl et al., who proposed a cut-off value of 2.0 K/µL as a predictor of grade 2 or higher irAEs [4,6,25,26]. These findings concerning the predictive value of ALC measured by routine blood tests may reflect intrinsic patient characteristics related to more specific cell subpopulations, in particular lymphoid cell subtypes. Indeed, patients with ipilimumab-mediated colitis tend to have a higher absolute count of peripheral CD4 T-cells and a lower percentage of regulatory T-cells at baseline than patients without colitis [27].

In addition, ANC values decreased after the first cycle of ICI, and low values of post-first ICI cycle ANC were a risk factor for experiencing an irAE in our cohort. A reduction in ANC after a first ICI cycle may be related to neutrophil depletion, a phenomenon that has previously been associated with improved clinical response to ICI therapy as a surrogate marker of T-cell hyperactivation [28]. It should be noted, however, that the granulocytic compartment detected by routine blood cell analysis includes both neutrophil- and granulocytic-myeloid-derived suppressor cells [29], including low-density neutrophils, which is a myeloid cell subset with immunosuppressive properties that has been associated with a poor response to pembrolizumab in non-small-cell lung cancer patients [30].

Consistent with the findings of a pilot study by our group [11], in the current cohort, women were more likely to develop an irAE than men. Sex differences in immune response have been extensively demonstrated [31]. Despite the widely accepted susceptibility of women to inflammatory and autoimmune diseases, it remains unclear whether there are sex differences in the incidence of irAEs [12], though it is known that women have a higher risk of severe irAEs than men [13]. Notably, in the current study, female sex showed an interaction in that the risk of an irAE increased further when female sex was combined with a low post-first ICI cycle ANC. Recently, it has been reported that women have higher absolute neutrophil counts than men, especially at younger ages, and that these differences decrease over the years [32]. The mechanism for this dimorphism may be mediated by estrogen-related hormonal or X-linked genetic factors [33,34]. Overall, our findings suggest a potential differentiating role of female sex in ICI-related toxicity, which could be mediated by certain blood cell populations, such as neutrophils, a hypothesis that should be investigated in future research.

Despite doubts about the impact of irAEs on overall survival [35,36], most studies have shown a positive correlation between irAE incidence and ICI response rate [37]. Generally, irAE-related mortality is associated with high-grade events [38]. Nonetheless, low-grade irAEs may herald more severe toxicities; therefore, in clinical practice, it is useful to be able to predict low-grade events. In our cohort, as many as 28.8% of patients experienced second irAEs, and a third of these were more serious than the first irAEs. In addition, some of these later irAEs involved key organs in the form of adrenalitis, encephalitis, or pneumonitis. Therefore, the clinical implementation of validated irAE prediction models based on baseline or early biomarkers could improve prognosis and quality of life in patients treated with ICIs.

Our study has several limitations that should be recognized. First, the severity and frequency of some irAE types were lower than previously reported [39]. Furthermore, pneumonitis was virtually absent, and there were no cases of central nervous system toxicity in our series. These findings may be partly explained by the design of our study, which focused on predictive biomarkers (i.e., factors present before irAE onset) and only included the first irAE for each patient, with subsequent irAEs being censored. As pulmonary and cerebral toxicities usually occur late in the course of ICI therapy, pneumonitis and encephalitis were not the first clinical manifestations in patients who developed irAEs, so both of these types of irAEs were under-represented in our study. Nonetheless, although we have not included these data in the statistical analysis, we have reported cases of both pneumonitis and encephalitis beyond the first irAE during follow-up (Appendix A). Second, the follow-up period was short for some patients due to high cancer-related mortality and not homogeneous for the overall cohort. Nevertheless, this limitation was managed using Fine and Gray competing risk models, which assessed the risk of developing an irAE over time in the presence of death as a competing risk. Finally, due to its pan-cancer design, this study did not provide data on tumor progression or cancer-related survival, which prevented us from performing a subgroup analysis of irAEs as a function of whether patients were responders or nonresponders to ICI therapy. Such subgroup analysis might help us to better understand the interaction between irAE occurrence and ICI response and the relationship between the predictive factors for both outcomes. Despite these drawbacks, we believe that the current study, from a real-life, multicenter, prospective, and pan-cancer cohort specifically designed to identify irAE predictors, provides practical insights into ICI-related toxicity.

## 5. Conclusions

Factors associated with irAEs in a cohort of cancer patients receiving ICIs were female sex, high pre-first ICI cycle ALC, and low post-first ICI cycle ANC. Pre-first ICI cycle ALC and post-first ICI cycle ANC seemed to impact the risk of irAEs in women but not in men. Further, regarding the prediction of irAEs, we identified an interaction between female sex and low post-first ICI cycle ANC. Priority should be given to the development of irAE prediction models and their validation in multiple settings, and such models should assess the role of sex differences in ICI-related toxicity.

## Figures and Tables

**Figure 1 cancers-16-00151-f001:**
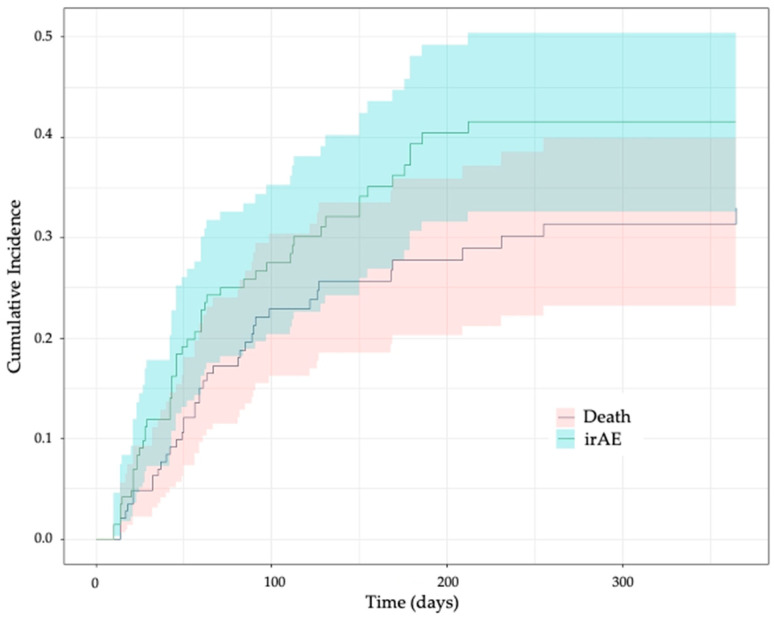
Cumulative incidence of immune-related adverse events over time in the presence of death as a competing risk. irAE, immune-related adverse event. The plot was generated using the Fine and Gray competing risk method. The colors show the 95% confidence band.

**Figure 2 cancers-16-00151-f002:**
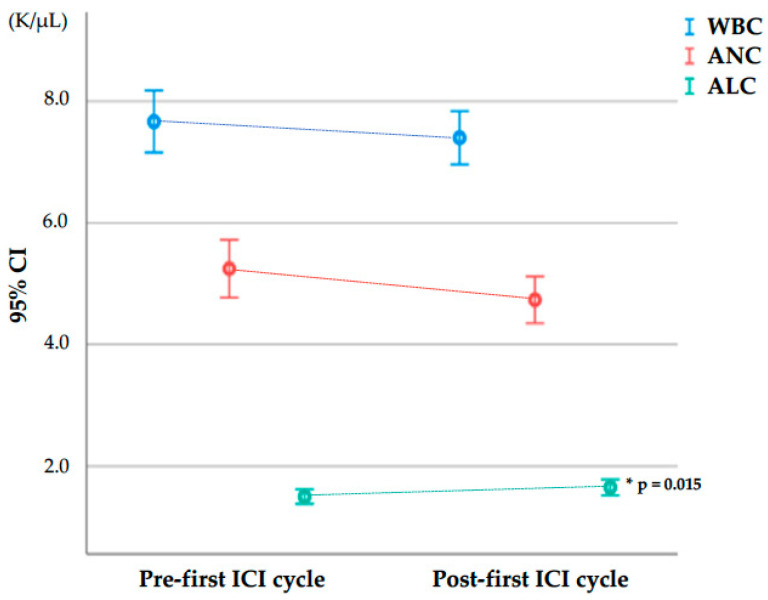
White blood cell, absolute neutrophil, and absolute lymphocyte counts at baseline (pre-first ICI cycle) and after the first ICI cycle (post-first ICI cycle) in the 134 patients who reached the second ICI cycle. Abbreviations in alphabetical order: ALC, absolute lymphocyte count; ANC, absolute neutrophil count; CI, confidence interval; ICI, immune checkpoint inhibitor; K/µL, thousand cells per microliter; WBC, white blood cell count. Bars represent 95% confidence intervals for the mean. * Comparisons between pre- and post-first ICI cycle using a Wilcoxon signed-rank test for paired data.

**Figure 3 cancers-16-00151-f003:**
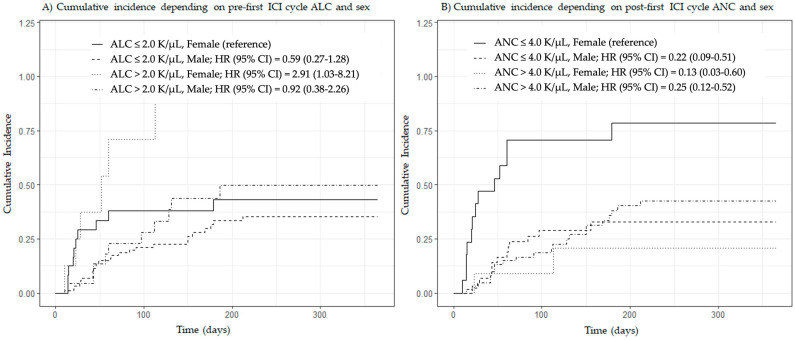
Cumulative incidence of immune-related adverse events by patient sex and predefined cut-off values of pre-first ICI cycle absolute lymphocyte count and post-first ICI cycle absolute neutrophil count. Abbreviations in alphabetical order: ALC, absolute lymphocyte count; ANC, absolute neutrophil count; CI, confidence interval; HR, hazard ratio; ICI, immune checkpoint inhibitor; K/µL, thousand cells per microliter. Plots were generated using the Fine and Gray competing risk method. Subgroups were created using cut-offs of 2.0 (**A**) and 4.0 (**B**) K/µL.

**Table 1 cancers-16-00151-t001:** Baseline characteristics of patients and treatments in the cohort (*n* = 145).

Variable	
Age at ICI initiation, mean ± SD	65.8 ± 9.7
Sex, *n* (%)	
Male	112 (77.2)
Female	33 (22.8)
ECOG score, *n* (%)	
0	41 (28.3)
1	89 (61.4)
2	15 (10.3)
Smoking history, *n* (%)	
Never smoker	31 (21.4)
Current or former smoker	114 (78.6)
Body mass index *, mean ± SD	26.1 ± 4.8
Renal failure ^†^, *n* (%)	8 (5.5)
Autoimmune disease, *n* (%)	9 (6.2)
Treatment type at ICI initiation, *n* (%)	
Adjuvant or neoadjuvant	9 (6.2)
First-line	77 (53.1)
Second-line	52 (35.9)
Third-line and beyond	7 (4.8)
Treatment regimen, *n* (%)	
Monotherapy	122 (84.1)
Pembrolizumab	45 (31.0)
Nivolumab	34 (23.4)
Atezolizumab	20 (13.8)
Durvalumab	11 (7.6)
Cemiplimab	6 (4.15)
Avelumab	6 (4.15)
Dual therapy (ipilimumab plus nivolumab)	23 (15.9)

Abbreviations in alphabetical order: ECOG, Eastern Cooperative Oncology Group; ICI, immune checkpoint inhibitor; SD, standard deviation. * Calculated as: body mass index = weight (kg)/(height (m))^2^. ^†^ Defined as a baseline glomerular filtration rate below 60 mL/min/m^2^.

**Table 2 cancers-16-00151-t002:** Summary of the first immune-related adverse event in patients of the cohort *.

irAE Type	*n* (%)
Cutaneous	17 (32.7)
Maculopapular rash	12 (23.1)
Pruritus	1 (1.9)
Other cutaneous irAEs **	4 (7.7)
Endocrine	14 (26.9)
Thyroiditis	3 (5.8)
Hyperthyroidism	5 (9.6)
Hypothyroidism	5 (9.6)
Adrenal insufficiency	1 (1.9)
Musculoskeletal	5 (9.6)
Arthromyalgia	2 (3.8)
Inflammatory arthritis	2 (3.8)
Polymyalgia rheumatica	1 (1.9)
Colitis	4 (7.7)
Pneumonitis	1 (1.9)
Nephritis	3 (5.8)
Hepatitis	2 (3.8)
Hematological irAEs ^†^	3 (5.8)
Miscellaneous ^‡^	3 (5.8)
Total	52 (100)

Abbreviations: irAE, immune-related adverse event. * In patients who experienced more than one irAE, only the first irAE was included in the analysis. ** Including two cases of psoriasiform rash and one case each of eczema and hypertrichosis. ^†^ Including one case each of hemolytic anemia, neutropenia, and pancytopenia (all recovered during follow-up). ^‡^ Including one case each of systemic lupus erythematosus flare, mucositis, and uveitis.

**Table 3 cancers-16-00151-t003:** Factors associated with occurrence of a first immune-related adverse event.

Univariate Analysis			
Variable	HR	95% CI	*p*-Value
Female sex	2.04	1.10–3.70	0.025
Dual therapy (ipilimumab plus nivolumab)	1.87	1.02–3.43	0.043
Pre-first ICI cycle ALC	1.60	1.11–2.31	0.011
Post-first ICI cycle ANC	0.81	0.68–0.95	0.012
**Multivariate Analysis**			
**Variable**	**HR**	**95% CI**	** *p* ** **-Value**
Female sex	2.17	1.20–3.85	0.010
Pre-first ICI cycle ALC	1.63	1.09–2.45	0.018
Post-first ICI cycle ANC	0.81	0.69–0.96	0.015

Abbreviations in alphabetical order: ALC, absolute lymphocyte count; ANC, absolute neutrophil count; CI, confidence interval; HR, hazard ratio; ICI, immune checkpoint inhibitor. Analyses performed using a Fine and Gray competing risk model with death as the competing event.

**Table 4 cancers-16-00151-t004:** Sex-stratified analysis of factors associated with occurrence of a first immune-related adverse event.

Sex-Stratified Model (Men)			
Variable	HR	95% CI	*p*-Value
Pre-first ICI cycle ALC	1.38	0.91–2.10	0.130
Post-first ICI cycle ANC	0.92	0.77–1.10	0.350
**Sex-Stratified Model (Women)**			
**Variable**	**HR**	**95% CI**	** *p* ** **-Value**
Pre-first ICI cycle ALC	2.61	1.40–4.86	0.003
Post-first ICI cycle ANC	0.57	0.41–0.81	0.002

Abbreviations in alphabetical order: ALC, absolute lymphocyte count; ANC, absolute neutrophil count; CI, confidence interval; HR, hazard ratio; ICI, immune checkpoint inhibitor. Analyses performed using a Fine and Gray competing risk model and considering death as the competing event.

## Data Availability

All data are available on request from the corresponding author. In addition, the data that support the findings of this study will be openly available in a publicly archived dataset, such as Mendeley Data or Zenodo.

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
