# Peer review of "Baseline Circulating Blood Cell Counts and Ratios and Changes Therein for Predicting Immune-Related Adverse Events during Immune Checkpoint Inhibitor Therapy: A Multicenter, Prospective, Observational, Pan-Cancer Cohort Study with a Gender Perspective"

_cancers, 2023, doi:10.3390/cancers16010151_

Round 1

Reviewer 1 Report

Comments and Suggestions for Authors

In this study, the risk of developing irAEs with immune checkpoint inhibitors across organs was assessed using information from blood tests.

As for the high incidence of mild irAEs, is the Grade of irAE at the onset or the worst? If it is the worst Grade, then I believe the evaluation should be done for Grade 2 or worse. It would be meaningful if Grade 1 irAE could predict more severe irAE, but of the 27 patients who experienced Grade 1 irAE, those who had Grade 2 or worse irAE Only four patients had Grade 2 or higher irAEs.

In addition, it is unknown whether the risk factors for concomitant use of CTLA4 antibodies are the same as those for no concomitant use of CTLA4 antibodies. Frequent irAEs differ with and without CTLA4 antibodies, and risk factors are expected to differ.

Gender was mentioned as a predictor of irAE, but is there a difference in the type of irAE by gender?

Author Response

Thank you very much for your comments. 
Please find attached the responses to your comments as a Word document. 

Reviewer 2 Report

Comments and Suggestions for Authors

 The work conducted by Teijeira might be promising, though the idea is not novel. However, this paper cannot be accepted in its current status. Several serious issues should be addressed and pronounced modifications should be added to the study design as follows:

1.     The authors should elaborate more in the introcuction on the different types of ICI, their uses, adverse effects. Please see point 3 below.

2.     The authors included any patient with solid organ cancer in the study. Would it be more precise to focus on certain types of solid tumors like lung or breast cancer?? This should be justified in the study.

3.     Also, the authors conducted the study for patients taking ICI therapy. It would be more accurate if the authors focused on certain types of ICI. The types of ICI are

-CTLA-4 (cytotoxic T lymphocyte-associated protein 4)

-PD-1 (programmed cell death protein 1)

-PD-L1 (programmed cell death ligand 1)

Each one is suitable for certain types of cancer, and the immune-related adverse events of

 each type is different from one another.

Thus, based on my comments 1 and 2, the authors should be more specific while conducting this case study, where patients with a specific type of tumor and being treated with a specific type of ICI  should be involved.

4.     More details on the patients' characteristics should be included in the study as there are several factors that would influence the findings of this study like age, type of carcinoma, histology (adeno, squamos, large cells, etc.), clinical stage , smoking status, line of immunotherapy (first, second or third), actionable mutation, type of immunotherapy (toripalimab, or which type is administered), Number of metastatic sites. All these factors should be considered while conducting this study.

5.     The paper is missing statistical interpretation in the discussion andd Figures. For instance, the current study's findings should be statistically sound compared to controls and previous studies. In Figures 1, and 3, please add means, Confidence intervals and probabilities.

Comments on the Quality of English Language

Minor

Author Response

(The authors gave the same response as above.)

Round 2

Reviewer 1 Report

Comments and Suggestions for Authors

Thanks for the additional analysis.

The data presented here does not go so far as to propose a model to predict irAE. The results of this study are limited in their usefulness to the reader. The authors should propose a model in the future.

Reviewer 2 Report

Comments and Suggestions for Authors

The authors responded to most of the raised concerns and I recommend publication in the current form.

Comments on the Quality of English Language

Minor edits.